# Integrative Analysis and Experimental Validation of Competing Endogenous RNAs in Obstructive Sleep Apnea

**DOI:** 10.3390/biom13040639

**Published:** 2023-04-01

**Authors:** Niannian Li, Yaxin Zhu, Feng Liu, Xiaoman Zhang, Yuenan Liu, Xiaoting Wang, Zhenfei Gao, Jian Guan, Shankai Yin

**Affiliations:** 1Department of Otolaryngology Head and Neck Surgery, Shanghai Jiao Tong University Affiliated Sixth People’s Hospital, Shanghai 200030, China; 2Shanghai Key Laboratory of Sleep Disordered Breathing, Shanghai Jiao Tong University Affiliated Sixth People’s Hospital, Shanghai 200030, China; 3Otolaryngology Institute, Shanghai Jiao Tong University Affiliated Sixth People’s Hospital, Shanghai 200030, China

**Keywords:** obstructive sleep apnea, competitive endogenous RNA, prediction model, inflammation, WGCNA, immune microenvironment

## Abstract

Background: Obstructive sleep apnea (OSA) is highly prevalent yet underdiagnosed. This study aimed to develop a predictive signature, as well as investigate competing endogenous RNAs (ceRNAs) and their potential functions in OSA. Methods: The GSE135917, GSE38792, and GSE75097 datasets were collected from the National Center for Biotechnology Information (NCBI) Gene Expression Omnibus (GEO) database. Weighted gene correlation network analysis (WGCNA) and differential expression analysis were used to identify OSA-specific mRNAs. Machine learning methods were applied to establish a prediction signature for OSA. Furthermore, several online tools were used to establish the lncRNA-mediated ceRNAs in OSA. The hub ceRNAs were screened using the cytoHubba and validated by real-time quantitative reverse transcription-polymerase chain reaction (qRT-PCR). Correlations between ceRNAs and the immune microenvironment of OSA were also investigated. Results: Two gene co-expression modules closely related to OSA and 30 OSA-specific mRNAs were obtained. They were significantly enriched in the antigen presentation and lipoprotein metabolic process categories. A signature that consisted of five mRNAs was established, which showed a good diagnostic performance in both independent datasets. A total of twelve lncRNA-mediated ceRNA regulatory pathways in OSA were proposed and validated, including three mRNAs, five miRNAs, and three lncRNAs. Of note, we found that upregulation of lncRNAs in ceRNAs could lead to activation of the nuclear factor kappa B (NF-κB) pathway. In addition, mRNAs in the ceRNAs were closely correlated to the increased infiltration level of effector memory of CD4 T cells and CD56^bright^ natural killer cells in OSA. Conclusions: In conclusion, our research opens new possibilities for diagnosis of OSA. The newly discovered lncRNA-mediated ceRNA networks and their links to inflammation and immunity may provide potential research spots for future studies.

## 1. Introduction

Obstructive sleep apnea (OSA), the second most common sleep disorder, is characterized by recurrent episodes of obstruction in the upper airway during sleep that in turn causes intermittent hypoxia (IH) and sleep fragmentation [1]. Epidemiological evidence showed that approximately over 936 million adults aged 30–69 years had OSA, and 425 million were estimated to be moderate to severely affected [2]. Men were substantially more likely than women to have moderate to severe OSA [3,4]. Patients with OSA always feel fatigued, have morning headaches, daytime sleepiness during the day, and dyslipidemia [5,6]. Without intervention, it causes multiple complications including cardiovascular disease [7], stroke [8], and Alzheimer’s disease [9], etc., which present huge socioeconomic burdens to both the individual and society as a whole.

Extensive studies have shown that IH, as a hallmark pathological feature of OSA, leads to oxidative stress, inflammasome activation, and hyperlipidemia, which contribute to the development and progression of OSA [10,11,12]. In addition, genetic factors together with certain types of environments can influence the development of OSA. Work by our lab and others has described the strong links between the gene polymorphisms of SLC52A3 [13], TNF-α [14], and the pathogenesis of OSA. However, OSA is a complex disease with polygenic inheritance, and single-gene mutations alone might not fully explain the genesis and development of OSA [15,16]. More efforts need to be focused on revealing the underlying mechanisms of OSA. So far, the nocturnal polysomnogram (PSG) [17] is the gold standard in OSA diagnosis. However, its use is limited due to the number of signals needed to track the PSG, the cost, the time taken and complexity, which renders early OSA diagnosis and timely intervention problematic. Therefore, cheaper, easier, and more accessible diagnostic testing would assist disease management and lower the societal toll of the disease.

Large parts of the human genome produce non-coding RNAs that lack protein-encoding capabilities such as microRNAs (miRNAs), long non-coding RNAs (lncRNAs), and various other classes [18]. Among them, miRNAs are transcripts with a length of 19–22 nucleotides in general, which post-transcriptionally regulate gene expression by binding to specific recognition sites known as miRNA recognition elements (MREs) on target transcripts [19].While lncRNAs exceed 200 nucleotides in length, they have emerged as important players in the regulation of gene transcription, splicing, and translation by directly combining with DNA, RNA, or proteins [20]. Accumulating evidence has demonstrated that ncRNAs play an important role in various pathological processes and diseases including OSA. LncRNA XIST was found to be upregulated and contributed to inflammation in the adenoids of patients with OSA [21]. Several lncRNAs could explain some of the underlying mechanisms of the cardiovascular damage caused by OSA [22,23,24]. Additionally, downregulated miR-21-5p was negatively correlated with the apnea–hypopnea index and oxygen desaturation index in OSA patients [25]. Fernando Santamaria-Martos and his colleagues showed that the circulating miRNA panel could be a potential biomarker for OSA diagnosis [26]. However, exploring the characterization of ncRNA molecules and their functional roles for OSA is still in its infancy, and further efforts are warranted to elucidate the mechanism underlying them.

Leonardo Salmena et al. proposed the “competing endogenous RNA” (ceRNA) hypothesis in 2011, and, since then, ceRNA-related research has developed rapidly and made great contributions to revealing the occurrence and development of diseases [27], prognosis [28], and therapeutic response [29]. The ceRNA hypothesis proposes that RNAs, including messenger RNAs (mRNAs) and ncRNAs, expressed concurrently and with a similar complement of MREs, are capable of indirectly regulating one another by competing for a shared, limited pool of miRNA molecules [30,31]. In other words, if one ceRNA (mRNA or lncRNA) increases, it titrates away miRNA from repressing other ones, and increases expression of another ceRNA (mRNA or lncRNA) in the network. To our knowledge, there are no studies to date examining the roles of ceRNA in OSA. Therefore, understanding ceRNA interaction in great depth may explain the mechanisms underlying OSA pathophysiology.

In this study, for the first time, we constructed the lncRNA-mediated triple regulatory networks in OSA. We compared the differential expression profiles between OSA and healthy controls collected from the NCBI GEO database. Weighted gene correlation network analysis (WGCNA) was performed to screen out key modules and mRNAs associated with OSA. Afterward, we applied machine learning methods to establish an mRNA signature, which possesses a good performance in classifying OSA patients and is healthy in both training and validation sets. Several online databases, Cytoscape software, and quantitative real-time reverse transcription-polymerase chain reaction (qRT-PCR) experiments were used to construct the ceRNA networks in OSA. We also explored the potential regulatory mechanisms of the ceRNA networks. Of note, hub ceRNA networks showed closely correlation with the inflammation response and changes in the immune microenvironment in OSA. We hope this study can provide new avenues for understanding the molecular mechanisms in OSA.

## 2. Materials and Methods

### 2.1. Data Collection and Preprocessing

The mRNA expression profiles of OSA, including GSE135917 [32], GSE38792 [33], and GSE75097 [34], were compiled from the GEO database using the R package “GEOquery” [35] (version 2.60.0). The GPL6244 (Affymetrix Human Gene 1.0 ST Array) platform was used for both GSE135917 and GSE38792 datasets. Studies have shown that continuous positive airway pressure (CPAP) can alter the expression profiles of OSA patients [36,37,38], even returning them to their initial, untreated levels [39]. We were therefore concerned that including the transcriptome of patients undergoing CPAP treatment may have masked changes that were strongly linked to the occurrence and development of OSA. So, in this study, we excluded 24 OSA patients who had followed CPAP treatment in GSE135917 and then merged it with GSE38792. The “removeBatchEffect” function of the limma [40] package (version 3.48.3) was used to remove the batch effect and ensure homogeneity of samples. A total of 16 normal controls and 44 OSA patients were included in the merged data. The GPL10904 (Illumina HumanHT-12 V4.0 expression beadchip) platform was used for the GSE75097. For our analysis, the “neqc” function of the limma package was used for background correction and normalization. We also excluded 14 OSA patients who had followed the CPAP analysis. At the end, 6 normal controls and 28 OSA patients were included as the validation set.

Next, we collected normalized data of the miRNA expression profile GSE99239 [41] from the GEO database. It used the GPL19128 (Exiqon miRCURY LNA microRNA array; 7th generation—hsa, mmu & rno (miRBase 18.0)) platform. GSE99239 consisted of 4 normal controls and 4 OSA patients, and all samples were included in the analysis.

The gene probes in the datasets included in this study were annotated with official gene symbols. The flowchart and details of data processing are shown in Figure 1.

### 2.2. Differential Expression Analysis

To identify differentially expressed genes (DEGs) between OSA patients and controls, the R package “limma” was implemented. Cut-off criteria were obtained for the merged mRNA expression data using adj *p*-value < 0.05 and |log2FC| > 0.58. Differentially expressed miRNAs (DEmiRNAs) were obtained by using adj *p*-value < 0.05 and |log2FC| > 2 as the threshold value. The results of differential expression analysis were displayed by using the R-package “ComplexHeatmap” [42] (version 2.8.0) and “ggplot2” [43] (version 3.3.5).

### 2.3. Weighted Gene Co-Expression Network Analysis

The weighted gene co-expression network analysis (WGCNA) is an algorithm that reveals gene co-expression modules, explores the relationship between gene modules and external clinical traits, and identifies genes that play key roles in diseases. In this study, we chose the top 25% of genes with the largest variations in expression profiles for WGCNA analysis by utilizing the “var” function and the “WGCNA” package [44] (version 1.70-3). The “hclust” function was used to cluster samples in the merged mRNA dataset and detect the outlier samples, and the sample GSM949170 was excluded. The “pickSoftThreshold” function was used to select the appropriated soft powers β to construct the scale-free topology network. Then, we constructed the gene correlations matrix, adjacency matrix, topological overlap matrix (TOM), and the corresponding dissimilarity (1-TOM) in turn. A hierarchical cluster tree was built according to the dissimilarity of the genes (distance = ”Euclidean distance”), and the dynamic tree-cut algorithm approach was applied using the “cutreeDynamic” function to determine the module of gene co-expression with values minBlockSize  =  50, and MergeCutHeight = 0.15. The module eigengene (ME) represented the expression profiles of each module, and the correlation between MEs and clinical traits of interest was calculated. Finally, module membership (MM) and gene significance (GS) were used to measure the correlation between a single gene and ME or clinical traits, respectively.

### 2.4. Functional Enrichment Analysis

To explore the underlying biological functions of the OSA-specific DEmRNAs, the “clusterprofiler” [45] package (version 4.0.5) in the R software was used to perform Gene Ontology (GO) and Kyoto Encyclopedia of Genes and Genomes (KEGG) analysis. With a threshold value of adj *p*-value < 0.05 (BH), the enriched GO terms, which included the biological processes (BPs), cellular components (CCs), and molecular functions (MFs), were visualized with the cnet plot, and the enriched KEGG pathways were visualized by using “pathview” [46] package (version 1.32.0).

### 2.5. Feature Selection, Modeling and Validation

The least absolute shrinkage and selection operator (LASSO) is a commonly used machine learning prediction method incorporating feature selection. By punishing the number of variables in the regression model, the coefficient of non-important variables can be reduced to 0 and then excluded. Random forest (RF) is an integrated machine learning method that has good robustness to noise data. It uses multiple decision trees for the joint prediction that can effectively improve the accuracy of the model. These machine learning methods were implemented with the R packages “glmnet” [47] (version 4.1-2) and “randomForest” [48] (version 4.6-14). We randomly divided the merged samples into a training set (80%) and validation set (20%), and the features with non-zero coefficients by the LASSO algorithm were maintained to generate the RF predictive signature. We validated the predictive performance of the signature using a 5-fold cross-validation. The R package “pROC” [49] (version 1.18.0) was used to plot the Receiver Operating Characteristic (ROC) curve and calculate AUC, which was used to evaluate the predictive accuracy. We also performed external validation in GSE75097 datasets.

### 2.6. Construction of the lncRNA-miRNA-mRNA Regulatory Networks

The lncRNA–miRNA–mRNA network was constructed based on the ceRNA theory following three steps. (1) The mRNA–miRNA interaction pairs: mRNAs that interact with DEmiRNAs were predicted by the miRTarBase database [16](http://mirtarbase.mbc.nctu.edu.tw, accessed on 24 December 2022, version 9.0), and subsequently intersected with OSA-specific DEmRNAs. Then, the overlapped mRNAs and corresponding miRNAs interaction pairs with negative correlations were kept. (2) The lncRNA–miRNA interaction pairs: interaction information of miRNA and lncRNA was extracted from the starBase database [50] (https://starbase.sysu.edu.cn/, accessed on 24 December 2022, version 2.0) and lncBase database [51] (http://www.microrna.gr/LncBase, accessed on 24 December 2022, version 2.0). To ensure the robustness of the interactions between miRNAs and lncRNAs, we took the intersection between these two databases. (3) A lncRNA–miRNA–mRNA regulatory network was built using a combination of mRNA–miRNA pairs and miRNA–lncRNA pairs with a continuous targeting relationship, and the network was mapped and visualized utilizing the Cytoscape software (version 3.8.2, National Institute of General Medical Sciences (NIGMS), Bethesda, Maryland, USA). Hub factors were pointed out using cytoHubba [52] (version 0.1), a plugin of the Cytoscape. We calculated the degree for each node in the ceRNA network through the degree topological algorithm. Hub factors were defined as the top 15 genes with the highest degrees, and their interaction networks were regarded as the hub ceRNA networks.

### 2.7. Correlation Analysis between Hub mRNAs and Immune Characteristics

Single sample gene set enrichment analysis (ssGSEA) relies on a backend deconvolution method that correlates particular cell types with a set of signature genes. The absolute enrichment degree of a certain immunocyte is reflected by calculating the enrichment fraction of each gene set in each sample. We obtained the immune cell infiltration gene set from the previous literature, and the “GSVA” package [53] (version 1.40.0) was used for ssGSEA analysis. Correlation between key genes and immune infiltrating cells was determined by Spearman correlation analysis and visualized with the “ggcor” package [54] (version 0.9.8.1).

### 2.8. Validation the Expression of the ceRNA In Vitro

#### 2.8.1. Cell Culture

HEK293T cells were purchased from the cell bank of the Shanghai Institute for Biological Sciences (CAS) and cultured in DMEM/high glucose medium with 10% FBS (Invitrogen, CA, USA), 100 U/mL penicillin, and 100 mg/mL streptomycin (Life Science, Washington, DC, USA). Cell lines were incubated in standard conditions (37 °C, 5% CO_2_) and tested negative for mycoplasma contamination.

#### 2.8.2. Chronic Intermittent Hypoxia Protocols

The intermittent hypoxia unit could switch the oxygen concentration in the cabin according to the settings by using an electronic gas flow meter and a computer-controlled valve. All units were kept at 37 °C in a conventional cell incubator with their gas supply tubes. In short, cells were divided into two groups: CIH and control. Then, cells were exposed to CIH conditions (the concentration of O_2_ lowered from 21% to 0% in 15 min, 5% CO_2_/95% N_2_ for 10 min, reoxygenation in 4 min, 21% O_2_/5% CO_2_/75% N_2_ for 1 min, and two anoxic cycles per hour) or normoxia conditions (21% O_2_/5% CO_2_/75% N_2_) for 48 h.

#### 2.8.3. RNA Extraction and qRT-PCR

Total RNA was extracted from 293T cells with Trizol Reagent (Invitrogen) according to the manufacturer’s instructions, and was reversely transcribed with RT reagent Kit gDNA Eraser (Takara, Kusatsu-shi, Japan). TB Green® Premix Ex TaqTM II (Takara) was applied to detect cDNA expression levels with β-ACTIN as the internal reference. In addition, total miRNA was isolated using the EasyPure® miRNA Kit (TransGen, Beijing, China) following the manufacturer’s protocol. The miRNA expression analysis was performed using TransScript^®^ Green miRNA Two-Step qRT-PCR SuperMix (TransGen) with U6 as the internal reference. Real-time PCR was performed on the LightCycler System 2.0 (Roche, Mannheim, Germany) at a temperature of 95 °C for 30 s, followed by 40 cycles with a temperature of 95 °C for 5 s, and 60 °C for 30 s. The primers are shown in Appendix A. All experiments were repeated three times and the relative gene expression was calculated by the 2^-ΔΔ^Ct method.

### 2.9. Statistical Analysis

Data are expressed as the mean ± SEM. Differentially expressed nodes in the hub ceRNAs between normal and CIH groups were compared using the Student’s *t*-test. GraphPad Prism software (version 9.0.0, Dotmatics, Boston, MA, USA) was used to generate graphics. All statistical analyses were performed using R software (version 4.0.2., Free Software Foundation, Boston, USA), and all analyses were conducted using two-tailed testing, with *p* < 0.05 being considered statistically significant.

## 3. Results

### 3.1. Differential Expression Analysis between Controls and OSA Patients

The flowchart of this article is shown in Figure 1. Using adj *p*-value < 0.05 and |log2FC| > 0.58 as the threshold, we identified 86 DEmRNAs between the controls and OSA patients, of which 59 mRNAs were upregulated and 27 were downregulated. The volcano plot and heatmap of the DEGs are shown in Figure 2A,C. Next, with adj *p*-value < 0.05 and |log2FC| > 2 as the screening threshold, we obtained 109 significantly expressed miRNAs between the controls and OSA patients, which consisted of 63 upregulated miRNAs and 46 downregulated miRNAs. The volcano plot and heatmap were carried out to illustrate the results of differential expression analysis (Figure 2B,D).

### 3.2. Identification of the Key Modules in OSA

The dendrogram was used to cluster samples in the merged mRNA dataset and detect the outlier samples (Figure 3A). In summary, a total of 44 OSA patients and 15 normal controls were included in the WGCNA analysis. We chose the soft-thresholding power β = 10 with scale-free R2 > 0.85 to establish a scale-free co-expression network (Figure 3B). Thirteen co-expression modules were identified through the hierarchical clustering method, and each module had over fifty genes in it (Figure 3C). Then, the correlation between modules and OSA was evaluated. The results showed that the brown (r = 0.53, *p* = 2 × 10^−5^), green–yellow (r = 0.48, *p* = 1 × 10^−4^), magenta (r = 0.53, *p* = 2 × 10^−5^), black (r = 0.35, *p* = 0.006), and yellow (r = 0.52, *p* = 3 × 10^−5^ ) modules were positively correlated with OSA, and the green module (r = −0.35, *p* = 0.006) was negatively correlated with OSA (Figure 3D). Among them, the brown module (r = 0.53, *p* =2 × 10^−5^) and magenta module (r = 0.53, *p* = 2 × 10^−5^), which included 568 genes and 526 genes, respectively, appeared to have the strongest link to OSA. Therefore, the brown and magenta modules were selected for further analysis. Correlations between module memberships and the gene significance for OSA are illustrated in the scatter diagram.

### 3.3. Functional Enrichment Analysis for OSA-Specific mRNAs

We intersected 86 DEmRNAs with genes in the selected brown and magenta modules, and, as a result, 30 OSA-specific DEmRNAs were obtained (Figure 4A). Subsequently, these overlapped genes were subjected to perform GO and KEGG analyses to investigate their potential biological functions. The results showed that GOBPs relating to antigen processing and presentation, lipoprotein metabolic process, and glycosylphosphatidylinositol (GPI) biosynthesis were significantly enriched (Figure 4B), indicating that immune and lipoprotein metabolic dysregulation is involved in the development of OSA. MF annotation revealed that the GOMFs were related to several enzyme activities (Figure 4C). Additionally, KEGG analysis suggested that the GPI-anchor biosynthesis pathway plays an important role in OSA.

### 3.4. Construction of the mRNA Signature and Validation

To explore whether OSA-specific mRNAs with the potential predictive value in predicting OSA, the machine learning methods were used to construct the predictive signature. LASSO was performed on the OSA-specific DEmRNAs for feature selection and dimension reduction, allowing irrelevant genes to be eliminated (Figure 5A,B). It was found that five DEmRNAs were crucial for OSA, including PTPN22, FAM200B, DYNLL1, FRZB, and TOMM22. Then, the RF signature was built based on these five genes with 1000 trees (Figure 5C). The %IncMSE and IncNodePurity indexes were calculated to measure the importance of variables. The results showed that PTPN22 was the most important variable in the model (Figure 5D). The ROC curve illustrated that the mRNA signature possessed a good performance in classifying OSA patients and was healthy in both the training set (AUC = 0.909) and validation set (AUC = 0.792) (Figure 5E,F).

### 3.5. Construct CeRNA Regulatory Networks

To identify mRNA–miRNA–lncRNA (ceRNA) networks in OSA, we used several online databases. A total of 6176 mRNAs predicted for 109 DEmiRNAs were acquired by using the miRTarBase database. Among them, 11 mRNAs and their interactions were selected based on the intersection with the 30 common DEmRNAs mentioned above. Furthermore, according to the theory of ceRNA, the candidate mRNA–miRNA interaction pairs in the ceRNA network should be negatively correlated. Therefore, we filtered the interaction network based on the expression profiles. From these, only 15 mRNA–miRNA pairs, including 6 DEmRNAs (upregulated) and 9 DEmiRNAs (downregulated), were left. These 9 DEmiRNAs were selected for further analysis, and the miRNA–lncRNA interaction pairs were obtained by the overlapped results of the prediction from starBase and lncBase databases (starBase database: 9 miRNA, 496 lncRNA, and 651 interaction pairs; lncBase database: 9 miRNAs, 109 lncRNAs and 165 interaction pairs), including 9 DEmiRNAs and 29 lncRNAs. Finally, the lncRNA-mediated ceRNA networks consisting of 6 DEmRNA nodes, 9 DEmiRNA nodes, 29 lncRNA nodes, and 100 interaction pairs were constructed (Figure 6A). We also visualized the chromosomal position of the nodes in the ceRNA regulatory network (Appendix A).

### 3.6. Identification of Hub CeRNA Networks

To trace hub ceRNA networks, we used cytoHubba, a plugin of Cytoscape software, to calculate degree values for each node. The top 15 identified hub factors included three mRNAs (OTUD4, RRN3, and ZNF117), seven miRNAs (hsa-miR-142-5p, hsa-miR-455-3p, hsa-miR-32-5p, hsa-miR-31-5p, hsa-miR-455-5p, hsa-miR-218-5p, and hsa-miR-3609) and five lncRNAs (KCNQ1OT1, NEAT1, XIST, OIP5-AS1, and ZNF561-AS1). Hub factors and their interactions are demonstrated in Figure 6B. The networks consist of 15 nodes and 30 edges, and different colors represent the degree of the hub factors. We further explored the expression level of the hub factors. As we can see, OTUD4, RRN3, and ZNF117 were upregulated in the OSA patients (Figure 7A–C), while hub miRNAs were all downregulated (Figure 7D–J). Since lncRNAs are based on predicted results, information relevant to expression was not available.

### 3.7. Validation of Expression Levels of Hub CeRNA Networks In Vitro

To verify the results of our bioinformatics analysis, we performed qRT-PCR. Cells treated with chronic intermittent hypoxia (CIH), which mimics the hypoxic condition during OSA, are the commonly used in vitro model for OSA (Figure 8A). As expected, compared with the control, significantly upregulated OTUD4, RRN3, and ZNF117 were observed in the CIH group (Figure 8B), while all the miRNAs in the ceRNA networks’ expression levels were significantly lower (Figure 8C). As for the lncRNAs, we found that KCNQ1OT1, XIST, and OPI5-AS1 were significantly upregulated, and only the ZNF561-AS1 was downregulated in the CIH group (Figure 8D), while lncRNA NEAT1 did not reach the significance level. According to the ceRNA theory, we removed ZNF561-AS1, NEAT1, and their interaction pairs. Finally, we proposed twelve ceRNA regulatory pathways in OSA, including four ZNF117-mediated (ZNF117-hsa-miR-455-3p-XIST, ZNF117-hsa-miR-455-3p-KCNQ1OT1, ZNF117- hsa-miR-455-5p-OIP5-AS1, and ZNF117-hsa-miR-455-5p-KCNQ1OT1), four RRN3-mediated (RRN3-hsa-miR-3609-KCNQ1OT1, RRN3-hsa-miR-32-5p-XIST, RRN3-hsa-miR-32-5p-OIP5-AS1, and RRN3-hsa-miR-32-5p-KCNQ1OT1), and four OTUD4-mediated (OTUD4-hsa-miR-455-3p-XIST, OTUD4-hsa-miR-455-3p-KCNQ1OT1, OTUD4-hsa-miR-31-5p-XIST, and OTUD4-hsa-miR-31-5p-KCNQ1OT1) ceRNA networks, respectively (Figure 8E). Their potential interaction sites are presented in Appendix A.

### 3.8. CeRNA Networks Related to Inflammation and Immune Characteristics of OSA

We further interrogated the potential molecular mechanisms of the ceRNA networks. Intriguingly, upregulation of all three lncRNAs in the ceRNA networks was closely correlated with cell apoptosis and the inflammatory response [55,56,57,58], as they had been reported to increase the expression level of NF-κB (Figure 9), indicating NF-κB pathway activation in OSA. GO analysis in this study suggested that OSA was closely correlated with altered immune function. Therefore, we performed correlation analysis for hub mRNAs with infiltrating immunocytes (Figure 10A). Of note, we found that hub mRNAs ZNF117 (r = 0.69, *p* = 1.4 × 10^−9^), RRN3(r = 0.67, *p* = 4.5 × 10^−9^), and OTUD4 (r = 0.6, *p* = 4.5 × 10^−7^) were most positively correlated with the abundance of effector memory CD4 T cells (Figure 10B–D). In addition, the level of immune infiltration of effector memory CD4 T cells in OSA was elevated (Figure 10E). In addition, OTUD4 (r = 0.56, *p* = 3 × 10^−6^) was significantly correlated with CD56^bright^ natural killer cell infiltration (Figure 10F). The number of CD56^bright^ natural killer cells was also higher in OSA patients (Figure 10G). The result of correlation analysis demonstrated that an increased number of memory CD4 T cells and CD56^bright^ natural killer cells infiltrated in OSA, which was closely influenced by the expression of mRNAs in the hub ceRNAs.

## 4. Discussion

It is estimated that a large number (> 80%) of adults with moderate-to-severe OSA remain undiagnosed [59]. Due to its serious complications, OSA drastically affects patient health and quality of life, as well as raises all-cause mortality. OSA diagnosis is still largely dependent on the clinical symptoms and nocturnal PSG, which leads to a low diagnostic rate of OSA. In the case of late diagnosis, first-line treatment for OSA such as nasal CPAP has limited effects on reversing neurocognitive damage [60]– [61,62]. Therefore, there is a need for novel screening methods that are inexpensive and capable of detecting OSA to reduce the burden on patients. However, it is a challenge since there is limited knowledge about the occurrence and development of OSA. Recently, the role of the genome and transcriptome in determining disease diagnosis and pathogenesis has become increasingly clear. NcRNAs have been shown to be involved in the occurrence and development of many diseases including OSA. Few studies, however, have examined whether ceRNA networks are related to OSA. Consequently, it is worthy of further study.

In this study, firstly, we systematically investigated the expression patterns of OSA patients. A total of 86 DEmRNAs and 109 DEmiRNAs were identified in normal controls and OSA patients. Meanwhile, we subjected an mRNA expression profile to WGCNA analysis, which can explore the key gene modules that are mostly related to the disease. Two modules with 1094 genes were most significantly associated with OSA. To make results more reliable, the OSA-specific mRNAs were defined as the intersecting mRNAs of these two approaches. Finally, 30 OSA-specific DEmRNAs were collected to further analysis.

To further evaluate the potential functions of the OSA-specific DEmRNAs, we performed the GO and KEGG analysis. The results showed that the “lipoprotein metabolic process” was significantly enriched. This is consistent with the previous points of view that OSA patients have dyslipidemia. Low-density lipoprotein cholesterol (LDL-C)/high-density lipoprotein cholesterol (HDL-C) increases in proportion to the severity of OSA, and may contribute partly to an increased risk for cardiovascular events in OSA patients [6]. In addition, CIH may be to blame for this aberrant lipid metabolism [12]. In addition, the “antigen processing and presentation” category was also enriched, corresponding to the previous results in the literature that “immunity and inflammation” were some of the largest biological processes upregulated in OSA [32]. The antigen presenting cells (APCs) turned out to be dendritic cells and macrophages, which play an important role in the immune response including innate and adaptive immunity. In fact, the polarization of macrophages and CD8+ T cells changed in CIH mouse models [63,64]. Moreover, Enrique Hernández-Jiménez found that hypoxic severity in OSA patients was linked to the polarization of monocytes and compromised activity of natural killer cells [65]. Min Sun and his colleagues showed that the monocyte-to-HDL-cholesterol ratio could be a marker of the presence and severity of obstructive sleep apnea in hypertensive patients [66]. More studies are needed to draw the map of the complex correlations between OSA and APCs in the future.

Next, five selected genes, including PTPN22, FAM200B, DYNLL1, FRZB, and TOMM22, were used to construct a signature through machine learning methods, which have good diagnostic accuracy in both cross-validation and external verification. Among them, a previous study showed that FRZB, which is involved in the signaling pathway, was downregulated in the placentae from women with obesity and obstructive sleep apnea [67]. However, there are no reports on the relationship between the other diagnostic genes and OSA. We hope that our research can provide directions for future experimental research revealing the potential mechanisms underlying OSA.

In the following part of our study, the lncRNA-mediated ceRNA networks were established according to the ceRNA theory, and qRT-PCR testing made our results more credible. A total of nineteen novel ceRNA pathways were identified, including four lncRNAs, six miRNAs, and three mRNAs. Although lncRNAs do not encode for proteins, they participate in various biological processes. LncRNA XIST was upregulated in the adenoids of OSA patients, which contributed to the inflammation by decreasing the expression of GRα and increasing the production of several inflammatory cytokines [21]. The relationship between OIP5-AS1 and cancer has been widely studied. It is plausible that it may play distinct roles (cancer-promoting or anti-cancer), depending on the types of cancers [68,69,70]. Experiments showed that upregulation of OIP5-AS1 was closely related to the acceleration of cell apoptosis and the inflammatory response [71,72]. NEAT1 is widely expressed in mammalian cell types, and is overexpressed in several inflammation-related disorders [73,74]. In the acute kidney injury mouse model, KCNQ1OT1 was highly expressed, and knockdown of the KCNQ1OT1 promoted cell proliferation and prevented apoptosis and inflammation [75]. NF-κB, a member of the Rel protein family, is present in almost all types of cells. Accumulating evidence has demonstrated its intricacy in regulating genes involved in cell growth and division, as well as apoptosis, hypoxia, stress, and the immune system. Interestingly, a large body of literature reports that upregulated expression of the lncRNAs in hub ceRNA networks causes the same event, i.e., upregulation of the NF-κB, through distinct molecular mechanisms [55,56,57,58]. As is well known, CIH in OSA patients can cause oxidative stress, systemic inflammation, and NF-κB-dependent inflammatory pathway activation [33,76], which is the potential pathogenesis of complications such as diabetes mellitus and cardiovascular disorders [77,78]. Based on these findings, we, therefore, speculated that the hub ceRNA networks may participate in the occurrence and development of OSA by regulating the activity of the NF-κB pathway and the inflammatory response. Further verification via experimentation and clinical studies is needed to support this hypothesis.

Inflammation and immunity are closely linked, and immune dysregulation in patients with OSA has been a matter of concern for decades. We explored whether the hub ceRNAs were associated with the alternation of the immunocyte infiltration in OSA. Of note, we found that OTUD4, RRN3, and ZNF117 in the ceRNAs were closely correlated with the infiltration memory CD4 T cells (r > 0.6, *p* < 0.05). In addition, OTUD4 was positively associated with the CD56^bright^ natural killer cells, suggesting they may play an important role in the development of immune dysregulation in OSA. Previous studies have shown that there is a significantly elevated infiltration of CD4+ T cells and NK cells, irrespective of age, in patients with OSA [79,80,81]. Elias A Said et al. suggested that the frequency CD4 T cells but not NK cells in OSA are associated with an increased expression of the nuclear protein Ki67 [82]. Distinct subtypes of immunocytes carry out different functions, and, though there have been numerous studies to unravel the changes of immune responses in OSA, few studies have examined sub-components of the immune system. Therefore, our study’s newly discovered lncRNA-mediated ceRNA networks and their links to inflammation and immunity may provide potential research spots for future studies.

To the best of our knowledge, this is the first time ceRNA networks have been introduced in OSA. However, we still have some limitations in this study. Firstly, although we carried out internal and external verification on the diagnostic model, the effectiveness of the diagnostic model must be confirmed in as many datasets as possible in the future. In this study, obviously, we used all the datasets available for OSA. Secondly, we confirmed the expression of the hub ceRNAs in the HEK293 cell lines using qRT-PCR, but there is still a need to confirm the results in multiple tissue-derived cell lines, animal experiments, or clinical patients. Additionally, more direct experimental support than that presented is still needed to determine whether the NF-κB pathway is involved in the mechanism of the ceRNA networks leading to OSA, and whether sub-immunocytes are changed in the OSA patients.

## 5. Conclusions

In this study, we evaluated the mRNA and miRNA expression patterns in OSA and explored the potential molecular mechanisms using bioinformatics. Furthermore, we established a signature using machine learning methods, which showed a good diagnostic performance for OSA. More importantly, we firstly constructed ten lncRNA–miRNA–mRNA regulatory networks and verified them in vitro. These may be involved in unrecognized mechanisms of inflammation and immunoregulation for OSA. Our findings provide fresh insights into the genetic and pathogenic mechanisms, laying the foundations for developing new diagnostic and therapeutic methods for OSA patients.

## Figures and Tables

**Figure 1 biomolecules-13-00639-f001:**
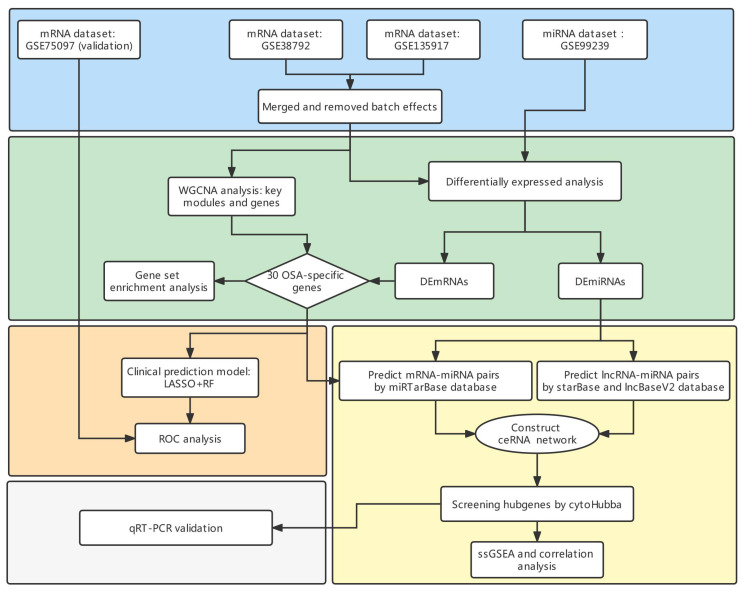
The flowchart of current study.

**Figure 2 biomolecules-13-00639-f002:**
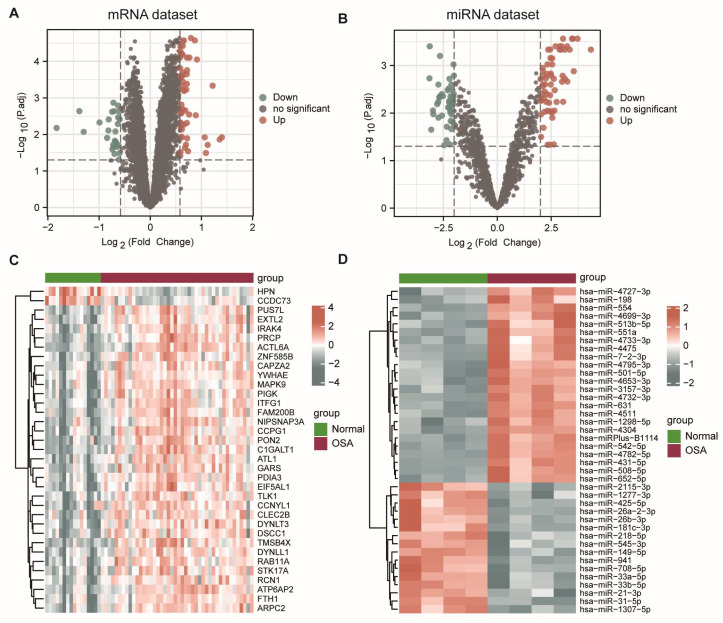
Volcano plots and heatmap plots of DEmRNAs and DEmiRNAs. (**A**) Volcano plot of DEmRNAs (126 controls vs. 119 OSA patients). The threshold set as the adj *p*-value < 0.05 and |log2FC| > 0.58. (**B**) Volcano plot of DEmiRNAs (4 controls vs. 4 OSA patients). The threshold set as the adj *p*-value < 0.05 and |log2FC| > 2. Green nodes indicate differences in OSA differentially downregulated, red nodes indicate mRNAs differentially upregulated in OSA. (**C**) Heatmap of the top 35 DEmRNAs with lowest adj *p*-value. (**D**) Heatmap of the top 35 DEmiRNAs with lowest adj *p*-value. Red means high expression, and blue means low expression. Each column represents a sample, and each row represents a gene.

**Figure 3 biomolecules-13-00639-f003:**
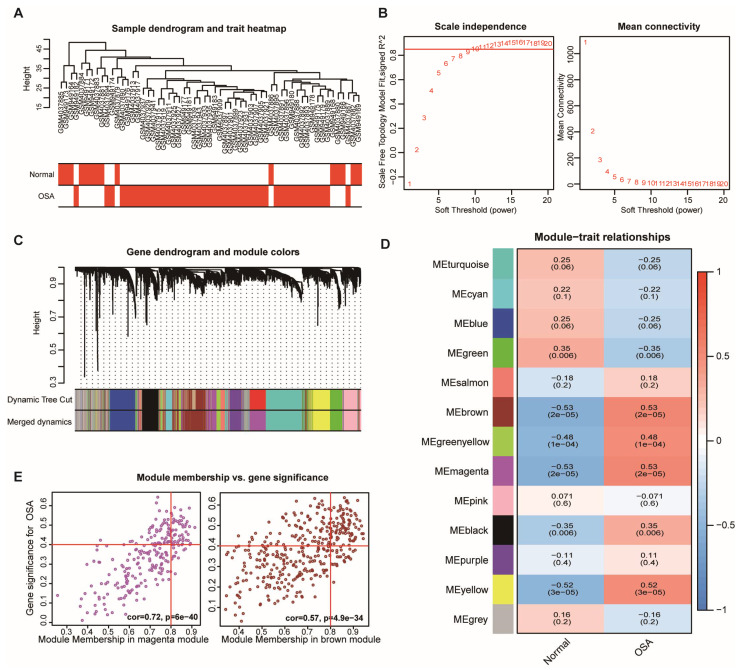
Weighted gene correlation network analysis for the mRNAs. (**A**) The dendrogram of sample and heatmap of the clinical trait. (**B**) Soft threshold powers analysis. The red line corresponds to 0.85. (**C**) Clustering dendrogram of the gene modules. Different colors represent different gene modules, and gray modules are made up of genes that do not belong to any of the modules. (**D**) The relationship between clinical traits and gene modules. Each cells contains the correlation coefficient (upper number) and the corresponding *p*-value (lower number). (**E**) The scatterplot of gene significance and module memberships in the magenta module (**left**) and the brown module (**right**).

**Figure 4 biomolecules-13-00639-f004:**
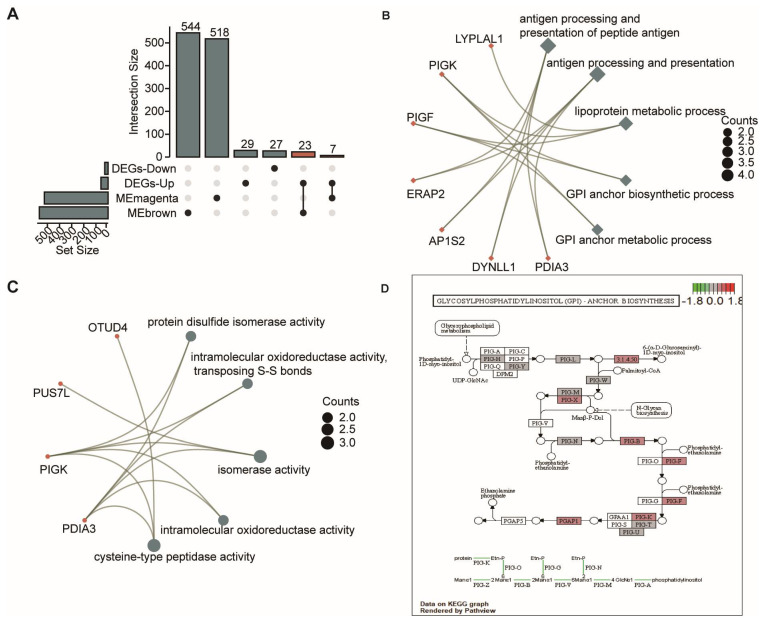
Identification of OSA-specific genes and enrichment analysis. (**A**)The Venn diagram of the genes in co-expression modules and DEmRNAs. (**B**) Significant enriched GO biological process terms. (**C**) Significant enriched GO molecular function terms. (**D**) KEGG pathway analysis and visualization of both gene expression and metabolomics data. Gene expression levels are indicated as significantly higher (red), unchanged (gray), or lower (green). The differences of hsa00563 glycosylphosphatidylinositol (GPI)-anchor biosynthesis pathways between OSA patients and normal controls.

**Figure 5 biomolecules-13-00639-f005:**
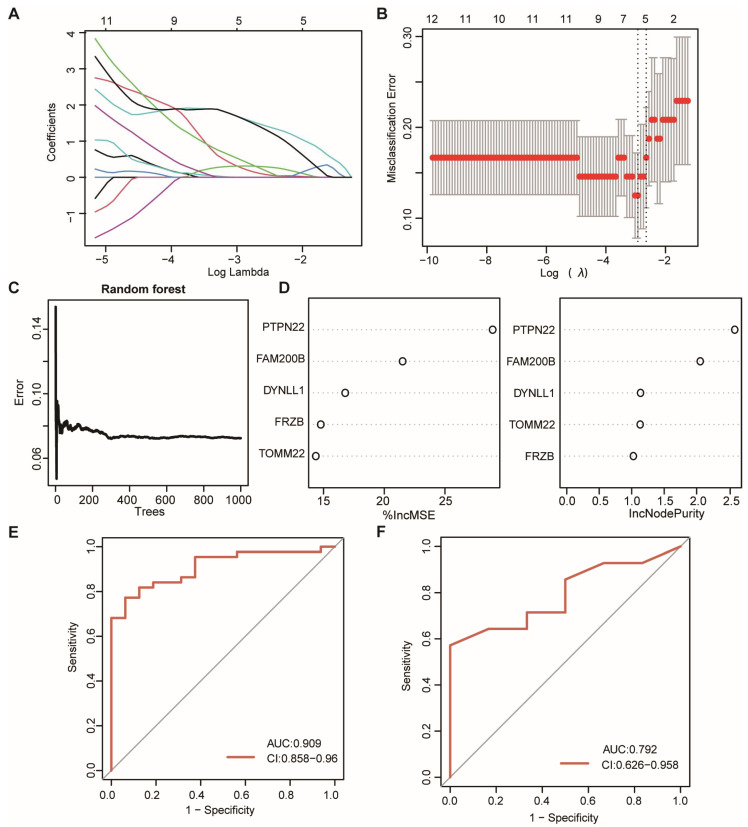
Construction of the predictive model for OSA. (**A**) LASSO coefficient profiles of OSA-specific genes. (**B**) 10-fold cross-validation for tuning parameter selection in the LASSO regression. (**C**) Correlations between the related errors and the number of decision trees. (**D**) The scatter plot of the RF variables based on the percentage of increase of mean square error (%IncMSE) (**left**) and the percentage of increase in node purity (%IncNodePurity) (**right**), respectively. (**E**, **F**) The prediction efficacy of RF model was evaluated by the ROC curve and AUC value based on the training set (**E**) and validation set (**F**).

**Figure 6 biomolecules-13-00639-f006:**
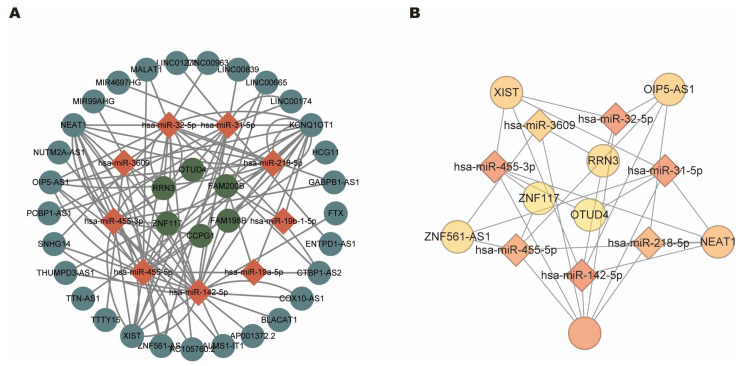
The competitive endogenous RNA regulatory networks. (**A**) The lncRNA-mediated ceRNA networks. (**B**) Visualization of the top 15 hub factors screened by the cytoHubba and their interaction networks.

**Figure 7 biomolecules-13-00639-f007:**
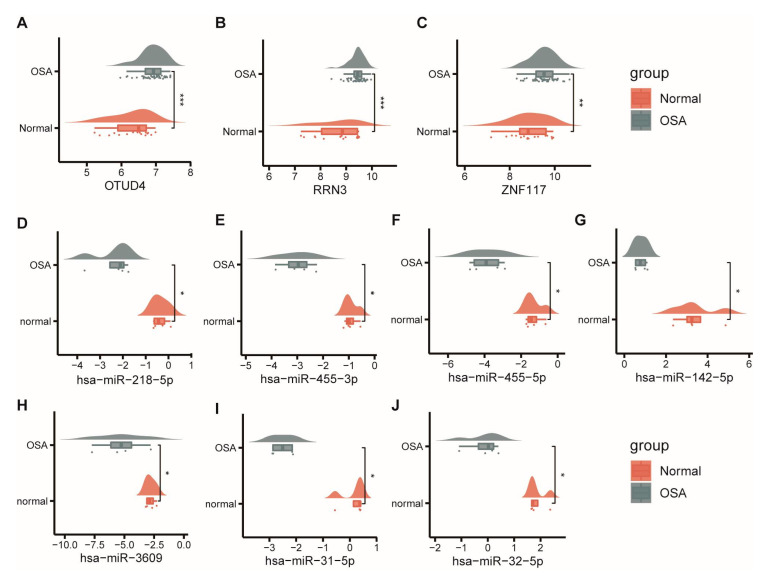
Expression analysis of the hub ceRNA networks. (**A**–**J**) The expression pattern of the nodes in the key ceRNA networks in OSA patients and normal controls. Mann–Whitney U test. * *p* < 0.05; ** *p* < 0.01; *** *p* < 0.001.

**Figure 8 biomolecules-13-00639-f008:**
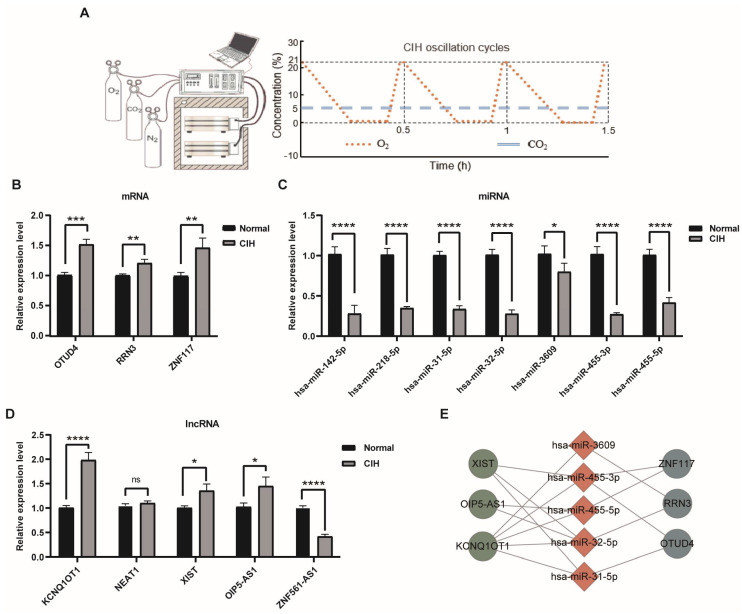
Validation of the hub ceRNA networks. (**A**) The schematic of the CIH condition. (**B**) Expression of the mRNAs in the hub ceRNA networks. (**C**) Expression of the miRNAs in the hub ceRNA networks. (**D**) Expression of the lncRNAs in the hub ceRNA networks. (**E**) The validated ceRNA networks. Unpaired Student’s *t*-test. * *p* < 0.05; ** *p* < 0.01; *** *p* < 0.001; **** *p* < 0.0001.

**Figure 9 biomolecules-13-00639-f009:**
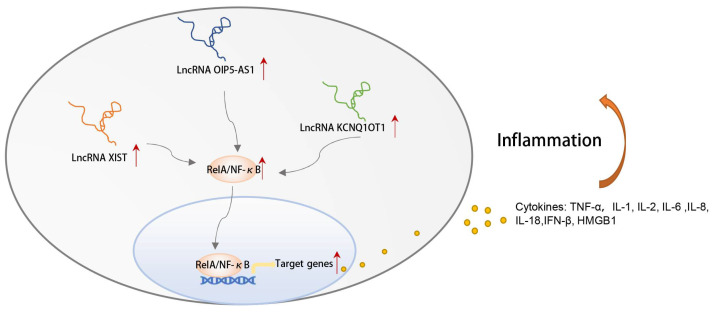
The sketch map of lncRNAs in the ceRNA networks activating the NF-κB signal pathway and inflammation based on the literature research. Red arrows indicate that the expression levels of the gene are elevated.

**Figure 10 biomolecules-13-00639-f010:**
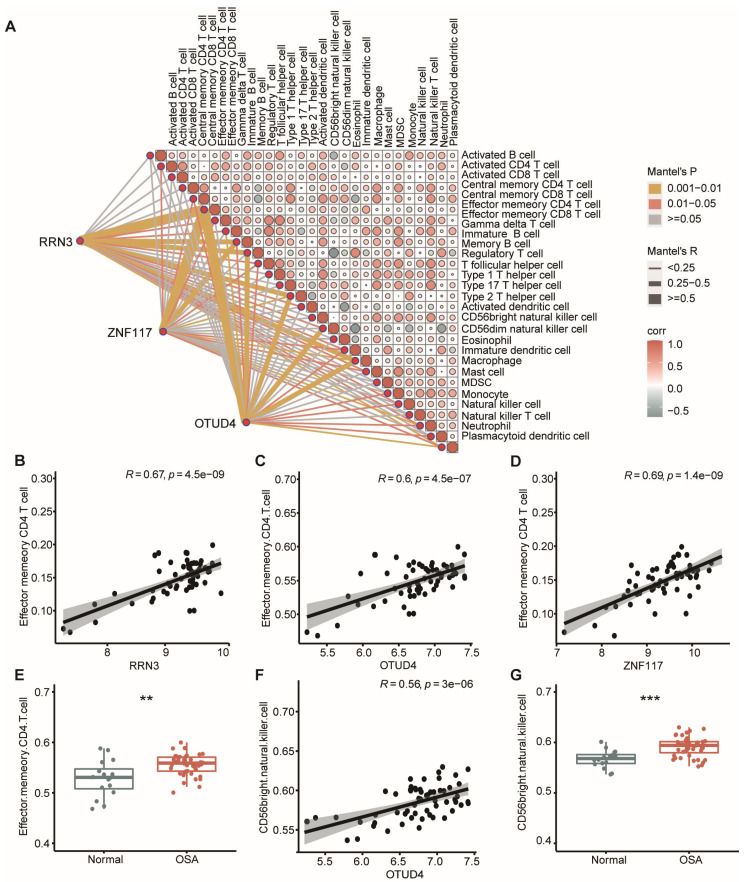
Relationship between hub mRNAs and the immune characteristics of OSA. (**A**) The complex correlation heatmap demonstrated correlations between hub mRNAs and the immunocytes as well as relationships between immunocytes. (**B**–**D**) Scatterplots of the correlations between RRN3 (**B**), OTUD4 (**C**), ZNF117 (**D**), and effector memory CD4 T cells. (**E**) The infiltration level of effector memory CD4 T cells in OSA patients and normal controls. (**F**) Scatterplots of the correlations between OTUD4 and CD56^bright^ natural killer cells. (**G**) The infiltration level of CD56^bright^ natural killer cells in OSA patients and normal controls. Mann–Whitney U test. * *p* < 0.05; ** *p* < 0.01; *** *p* < 0.001; **** *p* < 0.0001.

## Data Availability

The datasets presented in this study are openly available in online repositories. The names of the repository/repositories and accession number(s) can be found below: https://www.ncbi.nlm.nih.gov/geo/query/acc.cgi?acc=GSE135917/GSE38792/GSE75097/GSE99239 (accessed on 24 December 2022).

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
