# Peer review of "Integrative Analysis and Experimental Validation of Competing Endogenous RNAs in Obstructive Sleep Apnea"

_biomolecules, 2023, doi:10.3390/biom13040639_

Round 1
Reviewer 1 Report
Please consider attached document.

Reviewer 2 Report
The manuscript “Integrative Analysis and Experimental Validation of Competing Endogenous RNAs in Obstructive Sleep Apnea” by Li et al describes an interesting investigation into ceRNA in the pathology of OSA and its potential use as a molecular signature in the diagnosis of this disease. Research on ceRNAs in the regulation of gene expression is very recent and the study presented by Li et al is the first carried out on OSA.
The bioinformatics analyses based on databases and the in vitro validation study carried out are reliable, as well as the results obtained. For this reason, I recommend the publication of this manuscript in the presented form after a very minor revision as I suggest below.
Very minors
Abstract
Line 15 . “…NCBI GEO database” – please full name it as “National Center for Biotechnology Information (NCBI) Gene Expression Omnibus (GEO) database”
Introduction
Lane 91. “..Gene Expression Omnibus (GEO) database.” Now can be “…NCBI GEO database”
Results/Discussion
Line 292 “…prrdictive value..” - please correct to “…predictive value..”
In the Results section, please justify or comment on the use of HEK293 cells, a human kidney cell line, in the in vitro validation study. And/or in the Discussion section, mention the limitation, if any, of using this cell line in the validation study taking into consideration the disease under study.
Author contributions
Line 514 ..” ; X.Z. invistigation”- please correct to “ X.Z. investig
Reviewer 3 Report
This paper reported experimental results about ceRNAs in OSA. Various bioinformatics tools and statistical methods were used appropriately and effectively. The methods and the interesting results were sufficiently described. After some minor revision, it will be acceptable for publication.
[minor comments]
1. About clustering, it is simply mentioned as "hierarchical clustering" at line 252. If possible, I recommend the authors to describe the choices of distance (e.g. Euclidean) and linkage criterion (e.g. single-linkage). Additionally, I wonder whether the authors tried two or more different combinations of them because the right part of Figure 3F does not seems good. Isn't it possible to improve?
2. The authors used four datasets from GEO (GSE...), but did not compare their result and the results described in the papers of the datasets (e.g. PMID:31872261). Though I understand that the aim of the authors of this paper and the papers of the datasets are different, the author should compare the results if possible because some papers of the datasets also analyzed about DEGs and pathways (I think simple mentioning is sufficient).
line 85: "network.To" -> "network. To"
line 292: "prrdictive" -> "predictive"
line 346: "CytoHubba" -> "cytoHubba"
line 413: "ce-RNAs" -> "ceRNAs"
line 461: "rel protein" -> Isn't it "Rel protein"?
line 513: "conceptuualization" -> "conceptualization"
line 514: "invistigation" -> "investigation"
Round 2
Reviewer 1 Report
Concerns
1) The gap between text and citations. it should be consistent. See the citation in line 42 and 43.
2) Expand the word CPAP when used first time in the text. I think it is at line 111. Afterwards only abbreviations should be used. Wouldn’t it be useful to include this justification on CPAP treated patients in the supplementary material?
3) Point 2- I do not see any changes in the lines 116-120, as you mentioned in the response document.
4) Point 3- Have you included this description in the main text (or supplementary text)? I do not see any changes there.
5) Figure 2- Out of 86 DEmRNAs, how many were plotted in heatmap (Figure 2C)? Similarly for miRNA. Please mention. On what basis the genes/miRNA were selected to plot in heatmap?
6) Point 8- Places where these changes were made are not mentioned.
Round 3
Reviewer 1 Report
Thank you for addressing the suggestions.